# Polarization- and Chaos-Game-Based Fingerprinting of Molecular Targets of Listeria Monocytogenes Vaccine and Fully Virulent Strains

**Dmitry A. Zimnyakov** [1,2,*], **Marina V. Alonova** [1], **Maxim S. Lavrukhin** [2], **Anna M. Lyapina** [2] and **Valentina A. Feodorova** [2,3]

[1] Physics Department, Yury Gagarin State Technical University of Saratov, 77 Polytechnicheskaya Str., 410054 Saratov, Russia; alonova_marina@mail.ru

[2] Laboratory for Fundamental and Applied Research, Saratov State University of Genetics, Biotechnology and Engineering Named after N.I. Vavilov, 335 Sokolovaya Str., 410005 Saratov, Russia; lavrukhin.m.s@mail.ru (M.S.L.); lyapina_anna@inbox.ru (A.M.L.); feodorovav@mail.ru (V.A.F.)

[3] Department for Microbiology and Biotechnology, Saratov State University of Genetics, Biotechnology and Engineering Named after N.I. Vavilov, 335 Sokolovaya Str., 410005 Saratov, Russia

\* Correspondence: zimnykov@mail.ru

**Abstract:** Two approaches to the synthesis of 2D binary identifiers ("fingerprints") of DNA-associated symbol sequences are considered in this paper. One of these approaches is based on the simulation of polarization-dependent diffraction patterns formed by reading the modeled DNA-associated 2D phase-modulating structures with a coherent light beam. In this case, 2D binarized distributions of close-to-circular extreme polarization states are applied as fingerprints of analyzed nucleotide sequences. The second approach is based on the transformation of the DNA-associated chaos game representation (CGR) maps into finite-dimensional binary matrices. In both cases, the differences between the structures of the analyzed and reference symbol sequences are quantified by calculating the correlation coefficient of the synthesized binary matrices. A comparison of the approaches under consideration is carried out using symbol sequences corresponding to nucleotide sequences of the *hly* gene from the vaccine and wild-type strains of *Listeria monocytogenes* as the analyzed objects. These strains differ in terms of the number of substituted nucleotides in relation to the vaccine strain selected as a reference. The results of the performed analysis allow us to conclude that the identification of structural differences in the DNA-associated symbolic sequences is significantly more efficient when using the binary distributions of close-to-circular extreme polarization states. The approach given can be applicable for genetic differentiation immunized from vaccinated animals (DIVA).

**Keywords:** nucleotide sequences; identification; binary maps; polarization encoding; chaos game representation; *Listeria monocytogenes*; DIVA

## 1. Introduction

The essential part of modern bioinformatics is related to the quantitative analysis and visualization of quasi-random symbolic series obtained in the course of the DNA-sequencing procedure [1,2]. The explosive development of bioinformatics, which can perform essential simulations, in recent decades has given rise to an abundance of various approaches to the representation and analysis of linear nucleotide-based sequences in 2D, 3D, and higher-order spaces. The general idea of such representations is based on the creation of four-point bases and the step-by-step recurrent transformation of the analyzed linear symbolic (A, C, T, and G) sequence into an ensemble of mapping points in a higher-order space. Each point of the created basis is associated with one of the four basic nucleotides (adenine (A), cytosine (C), thymine (T), and guanine (G)).

A number of approaches to the visualization of DNA-associated (A, C, T, and G) sequences developed in the last two decades involve a recurrent synthesis of the representing

polylines in two or three dimensions, such as the H-line [3], C-line [4,5], Z-line [6], RY-, MR-, and WS-lines [7], etc. In these procedures, an appropriate choice of the basic configuration (namely, the establishment of certain associations between the base nucleotides and opposite points of the basis) makes it possible to additionally reveal some biochemical features of the analyzed DNA fragments. These features are the predominance of the keto groups over amino groups (or vice versa, the RY basis), purines over pyrimidines (or vice versa, the MK basis), and strong hydrogen bonds over weak bonds (or vice versa, the WS basis) in the sequences. They can be identified by analyzing the polyline projections onto the corresponding coordinate axes of the chosen basis [7].

An alternative approach to the polyline representation of nucleotide sequences, to a certain extent, is their two-dimensional display as dot sets on a plane region (usually square in shape). The most popular method using this dot-based displaying technique is the chaos game representation (CGR), introduced into bioinformatics by H.J. Jeffrey in 1990 [8]. Since that time, the CGR technique has been successfully used to analyze and visualize various nucleotide sequences many times (see, e.g., [9–13]). It should be noted that the general problem when using polyline and dot-based displaying techniques (including the CGR technique) for nucleotide sequences is the low level of their representations of relatively short fragments of nucleotide sequences (those with several hundred to several thousand bases). In this case, the emerging relatively low volume or surface density of the displayed line segments or dots does not allow, in particular, the confident identification of the presence of fractal properties in the displayed nucleotide structures, which is associated with existence of large-scale correlations of base positions in the sequences. From the point of view of the convenience of analyzing the resulting representations with relatively small lengths of displayed sequences, it is proper to fragment the display space onto a set of voxels and count the number of display elements in each voxel. In the case of chaos game representation, this concept is implemented via the so-called frequency chaos game representation technique (FCGR) [14–17].

As a specific approach to the two-dimensional representation of nucleotide sequences of a finite length, the polarization imaging technique [18,19] should also be noted. This technique is based on modeling the reading of a DNA-associated two-dimensional phase object (quasi-random phase screen) using a coherent light beam with a given polarization state and the subsequent analysis of the local polarization structure of the readout beam in the far diffraction zone. The DNA-associated quasi-random phase screen is a $2N \times 2N$ matrix, where $N^2$ is the number of nucleotide triplets in the displayed fragment of the nucleotide sequence. Accordingly, each triplet is associated with a certain $2 \times 2$ submatrix in the total $2N \times 2N$ matrix, and positions of the submatrix elements are associated with four basic nucleotides (e.g., the first element of the first row of a submatrix is associated with adenine and the second one with cytosine, the first element of the second row corresponds to thymine, and the last element of the submatrix is associated with guanine). The submatrix elements can take one of four values (0, 1, 2, or 3), with the value determined by the content of the corresponding nucleotide in the triplet coded by the submatrix; accordingly, the sum of all elements of any submatrix is always equal to 3. The values of the elements determine the phase modulation depth for the x- and y-polarized components of the readout coherent beam. In accordance with the phase modulation scheme proposed in [19], the values of the phase shifts for orthogonally polarized components of the readout beam passing through various elements of the $2N \times 2N$ phase screen can vary in the range of 0 to $3\pi/2$, with a step of $\pi/2$. The polarization states with the phase shifts of 0 and $2\pi$ are identical. This type of phase modulation leads to a high probability of the occurrence of close-to-circular local polarization states of the readout light field in the far diffraction zone. In turn, recovery of the spatial distributions of these extreme states, which exceed the specified level of discrimination, makes it possible to create a 2D binary structure, which in fact is a unique identifier ("fingerprint") of the analyzed nucleotide sequence. Such uniqueness follows from a rather rigid condition for the formation of a local circular polarization state in the far diffraction zone (equality of amplitudes of the superposing x- and y-polarized components

of the diffracted readout beam and the phase shift between them, exactly equal to $\pi/2$ or $3\pi/2$). In particular, a pilot verification of this methodology using model data and nucleotide sequences for three strains of the SARS-CoV-2 virus showed the high sensitivity of the synthesized binary maps to single-nucleotide substitutions in the analyzed sequences relative to the reference ones [19]. It should be noted that synthesis of 2D binary maps of extreme polarization states, in contrast to a vast majority of popular representation techniques, directly leads to the formation of identification matrices of 2N × 2N size, and does not require the additional fragmentation of the representation space and the counting of the imaging elements in individual voxels. This is a direct consequence of the inherent properties of the discrete Fourier transform underlying the polarization mapping technique.

From a biological point of view, the described computational methods may be applicable in the analysis of and discrimination between different strains of microorganisms that vary in terms of the number of substituted nucleotides, which circulate in certain populations. This task is of great importance for disease outbreak handling, for example, in farm animals, where the rapid differentiation of the circulating strains in vaccinated herds is critical at the beginning of the outbreak, especially in the case of attenuated vaccines, where a live pathogen replicates, stimulating the immune system of the host. For this reason, a genetic variant of "differentiate immunized from vaccinated animals" (DIVA) strategy may be employed [20]. Initially, the DIVA strategy proposed by J.T. van Oirschot in the early 1990s [21] implied the use of marker vaccines based on deletion mutants of wild-type microorganisms depleted of some immunogenic components and the subsequent implementation of accompanying serological tests to determine the antibody response against the antigens lacking in the vaccine strain. This approach enables a differentiation between the vaccine-induced and post-infection immune responses [22]. However, the classical DIVA strategy requires time for antibody accumulation, and is not very useful when the elicited specific humoral immune response is not pronounced. In contrast, DIVA PCR-based tests detect genetic differences between the vaccine and wild type field strains.

Listeriosis is a severe infectious disease of humans and wild and farm animals with a high mortality rate of 30–50%, depending on the clinical form of the disease. The major causative agent is *Listeria monocytogenes*, a facultative intracellular bacterium capable of switching from a saprophyte to a parasitic lifestyle. *L. monocytogenes* is widespread in different ecological niches, and can successfully replicate in food products, which are the main source of infection for humans. Large foodborne outbreaks of listeriosis in humans often have parallels to outbreaks in animals [23]. Therefore, control of the infection in herds is one of the ways to control the human infection.

A number of vaccines against listeriosis are being developed for the prevention of the disease. Among them, live attenuated vaccines are considered as more potent because of they mimic natural infection and elicit the cell-mediated protective immune response needed against intracellular pathogens [24], while inactivated vaccines require adjuvants or special processing to induce a cellular response [25–28]. Natural or artificial attenuation is commonly due to the lack of mutations or mutations in the target genes coding the pathogen's virulence factors. Listeriolysin (LLO), a pore-forming toxin, is a major virulence determinant of *Listeria* which plays a crucial role in the intracellular lifestyle of pathogenic *Listeria* spp. LLO enables bacteria to escape from phagosomes and participates in cell-to-cell spread [29,30]. Substitution mutations in the structural *hly* gene coding LLO lead to a significant reduction in the virulence of mutated strains, providing a basis for the development of attenuated listeria vaccines [31]. LLO is also a highly immunogenic molecule, and induces innate and adaptive immune responses in different animal models and in humans [29,32,33]. LLO acts as a vaccine target, and immunization with a non-pore-forming LLO variant supplemented with the choler toxin as an adjuvant elicits protective T-cellular and humoral response in mice [34].

The purpose of this work is a comparative analysis of two approaches to the two-dimensional binary mapping of nucleotide sequences in terms of a unique correspondence between the structure of the synthesized maps and structure of the sequences (testing

the "fingerprinting efficiency"). The first approach is based on the consideration of 2D binary maps of extreme local polarization states. Accordingly, as the impact parameters, the discrimination threshold of local polarization states and the transformation scale of the DNA-associated phase screen into the analyzed polarization-dependent diffraction field are considered. The second approach is based on the use of a FCGR technique with assessment of the affect in the choice of RY-, MK- and WS-basis and degree of fragmentation (the voxel size) on the "fingerprinting efficiency"). Symbol series corresponding to nucleotide sequences of the *hly* gene of the vaccine and wild type *L. monocytogenes* strains differing in the number of mutationally substituted nucleotides were used as the model objects of analysis. In order to analyze the efficiency of the proposed approaches for nucleotide sequences with small numbers of substitutions, the modification of the original target sequences of the *L. monocytogenes* strains was carried out. This modification was performed using the procedure of the reverse translation of protein sequences, and the artificially generated sequences with reduced numbers of substitutions were also used for the analysis.

## 2. Materials and Methods

### 2.1. Analyzed Nucleotide Sequences

Nucleotide sequences of the *hly* gene encoded LLO in the *L. monocytogenes* strains, which were isolated from farm animals with the complete whole genome status and were accessed from the NCBI GenBank and used for the polarization- and FCGR-based binary mapping ("fingerprinting"). The list of the *L. monocytogenes* strains included the following:

1.  *L. monocytogenes* strain AUF, NCBI Acc. # NZ_CP048400.1. This is a commercial attenuated strain used for a live whole cell vaccine against listeriosis of farm animals in the Russian Federation. The parent virulent *L. monocytogenes* strain "A" was originally isolated in 1965 from the ovine brain specimen, which was derived from an ewe with neurolisteriosis in Novosibirsk Region. The vaccine strain AUF was obtained in the middle of 1960s under repeated UV treatment of the parent strain, and was then animalized through white mice [35–37]. The *hly* gene nucleotide sequence of the *L. monocytogenes* strain AUF was used as the original reference sequence.
2.  *L. monocytogenes* strain FSL-J1-158, NCBI Acc. # NZ_CP090057.1. This strain was originally isolated from caprine vaginal flushes in 1997 in the USA [38].
3.  *L. monocytogenes* strain FDAARGOS_607, NCBI Acc. # NZ_CP041014.1. This is the FDA standard strain isolated from sheep in 2019 in the USA.
4.  *L. monocytogenes* strain NTSN, NCBI Acc. # NZ_CP009897.1. This was isolated from sheep in 2011 in China [39].
5.  *L. monocytogenes* strain UKVDL9, NCBI Acc. # NZ_CP065028.1. This was isolated from the brain tissues of a sheep in 2014 in the USA [40].
6.  *L. monocytogenes* strain UKVDL4, NCBI Acc. # NZ_CP076644.1. This was isolated from cow liver in 2014 in the USA [40].
7.  *L. monocytogenes* strain 4/52-1953, NCBI Acc. # NZ_CP048401.1. This is the oldest strain isolated from a piglet in the current territory of Russian Federation in 1953 in the USSR [41].
8.  *L. monocytogenes* strain UKVDL7, NCBI Acc. # NZ_CP076669.1. This was isolated from horse liver in 2014 in the USA [40].

As was shown in the previous study [19], the polarization-based binary mapping approach displays a high sensitivity to a small number of substitutions (less than 10) in the structure of a pair of nucleotide sequences. The sensitivity decreases with an increasing number of differences. Thus, two sets of model data were used in this work to consider the fingerprinting techniques in a wide range (from ten to several tens) of substitutions. The first dataset (dataset # 1) is referenced above. The second model set with reduced numbers of substitutions includes the artificial sequences generated on the base of the first set using the EMBOSS Backtranseq tool (https://www.ebi.ac.uk/Tools/st/emboss_backtranseq/ (accessed on 25 September 2023)). The generation procedure is based on reverse translation of protein sequences as the input data, and results in the artificial nucleic acid sequences,

which represent the most likely non-degenerate coding sequences. Thus, the "simplistic" *hly* gene sequences (the dataset # 2) with a reduced number of substitutions regarding the reference AUF sequence # 1 (6–11 SNPs vs. 44–56 SNPs in the case of dataset # 1) were obtained. Note that the artificial sequences are one triplet shorter than the original sequences (1587 vs. 1590 base nucleotides) due to loss of the terminate codon. A comparison of the used datasets # 1 and # 2 is given in the Supplementary Materials, including the multiple sequence alignments of the original and artificial sequences (see Figure S1a,b and Tables S1 and S2).

With regard to polarization-based binary mapping for all eight model strains from the datasets ## 1 and 2, the relevant sequence fragments containing 1587 base nucleotides (with 529 triplets) were carefully considered. A preliminary frequency analysis of the combinations of basic nucleotides (adenine (A), thymine (T), cytosine (C), and guanine (G)) in the arbitrarily chosen triplets showed the significant inhomogeneity of the relative frequencies of their combinations. For example, Table 1 displays relative frequencies for the AUF artificial reference sequence from the dataset # 2. Note that the "n elements" designation determines the content of a given nucleotide in a triplet; accordingly, the last column in Table 1 gives the probabilities of finding triplets such as AAA, TTT, CCC, and GGG (these probabilities occur zero-equal for the finite-length sequence, except for the case of CCC).

**Table 1.** Relative frequencies of nucleotide combination occurrence in arbitrarily chosen triplets for the artificial AUF sequence (strain # 1 from dataset # 2).

|  | 0 Elements | 1 Element | 2 Elements | 3 Elements |
|---|---|---|---|---|
| A | 0.091 | 0.11 | 0.0487 | 0 |
| C | 0.07 | 0.132 | 0.036 | 0.012 |
| T | 0.171 | 0.073 | 0.0066 | 0 |
| G | 0.09 | 0.103 | 0.057 | 0 |

Table 2 displays the differences in the triplets between the A, C, T, and G sequences for dataset # 2; it can be seen that the maximal difference in the triplets occurs for the strains 1 and 2, and 8.

**Table 2.** Differences in the triplets between the artificial nucleotide sequences ## 1–8 (dataset # 2).

| The Triplet Position from the Start Codon | 1 | 2 | 3 | 4 | 5 | 6 | 7 | 8 |
|---|---|---|---|---|---|---|---|---|
| 14 | GTG | ATC | GTG | GTG | ATC | ATC | ATC | ATC |
| 31 | AAC | CAG | AAC | AAC | CAC | CAC | AAC | CAC |
| 34 | AAC | GAC | AAC | AAC | GAC | GAC | AAC | GAC |
| 35 | AGC | AGC | CTG | CTG | CTG | CTG | AGC | CTG |
| 309 | AAC | ACC | AAC | AAC | AAC | AAC | AAC | AAC |
| 438 | GTG | ATC | ATC | ATC | ATC | ATC | ATC | ATC |
| 523 | AAG | AGC | AGC | AGC | AGC | AGC | AGC | AGC |

*2.2. Methodology of Polarization-Based Encoding and Recovery of the Binary Maps of Extreme Polarization States*

In accordance with [19], the following stages of polarization-based encoding and mapping can be outlined:

- Synthesis of the virtual 2N × 2N DNA-associated quasi-random phase screen;
- Simulation of the polarization-dependent far-zone diffraction pattern formation due to reading out the synthesized phase screen by a coherent collimated beam with a given polarization state;

-     Selection of close-to-circular local polarization states in the simulated diffraction pattern (the latter procedure is carried out using the discrimination procedure for a spatial distribution of the fourth component of Stokes vector).

Each triplet in the displayed DNA-associated nucleotide sequence is encoded by a four-element squared matrix $\left(a_{ij}\right)_{2\times2}$ (submatrix) in accordance with the following rule: $\tilde{a}_{11} \to A, \tilde{a}_{1,2} \to C, \tilde{a}_{21} \to T, \tilde{a}_{22} \to G$ (note that establishment of the correspondences between the submatrix elements and basic nucleotides is arbitrary). The value of the corresponding element determines the content of the given nucleotide in the triplet. For the above correspondence rule, the encoding examples are as follows:

$$\begin{pmatrix} 2 & 0 \\ 1 & 0 \end{pmatrix} \to AAT; \begin{pmatrix} 0 & 0 \\ 3 & 0 \end{pmatrix} \to TTT; \begin{pmatrix} 0 & 1 \\ 1 & 1 \end{pmatrix} \to CTG. \tag{1}$$

After representation of the nucleotide triplets by a set of submatrices, the main phase-modulating matrix $\left(a_{ij}\right)_{2N\times2N}$ (the virtual phase screen) is synthesized through a line-by-line assembling of the submatrices in accordance with the order of triplets in the analyzed sequence. The size of the analyzed sequence fragment must be equal to $N^2$.

Let us assume that the synthesized phase screen is read by a collimated linearly polarized light beam, with the polarization plane forming a 45° angle with the sides of the screen (Figure 1).

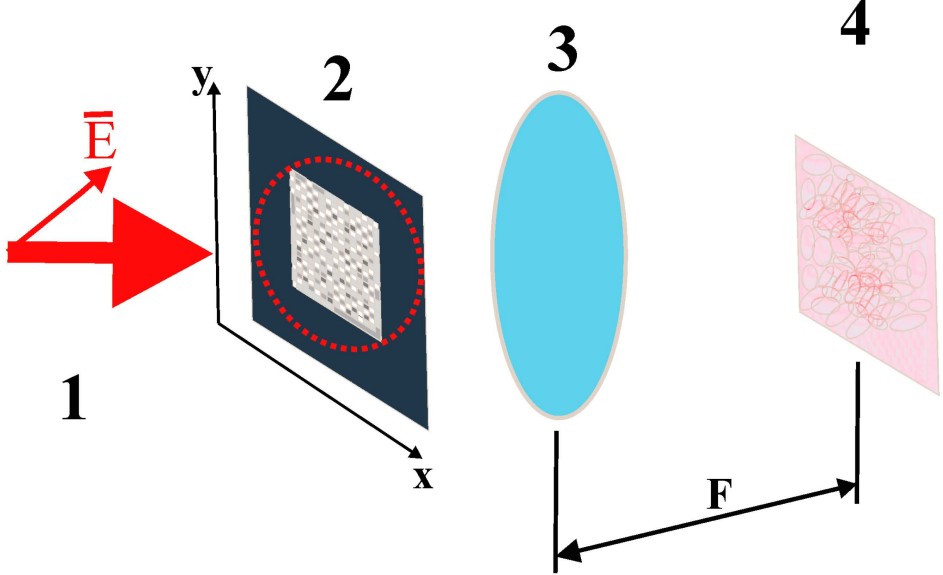

**Figure 1.** Diagram illustrating the principle of polarization coding and analysis of DNA-associated symbol sequences; 1—reading linearly polarized laser beam (the polarization plane is inclined at a 45° angle to the coordinate axes (x, y); 2—DNA-associated phase screen (the illuminated area is shown by a dashed red circle); 3—Fourier-transforming lens; 4—readout plane (a polarization-discriminating unit placed between the items 3 and 4 is not shown).

The phase modulation of the x- and y-polarized beam components is carried out in accordance with the following rule:

$$\left(\Delta\varphi_{ij}\right)^{x}_{2N\times2N} = 0;$$
$$\left(\Delta\varphi_{ij}\right)^{y}_{2N\times2N} = \left(\frac{\pi}{2}\right) \cdot \left(a_{ij}\right)_{2n\times2n}. \tag{2}$$

The amplitude distribution of orthogonally polarized components of the diffracted readout beam in the far diffraction zone can be described by the following expression (see, e.g., [42]):

$$E_{k,m}^{x,y} = \frac{1}{4N^2} \sum_{i=-N}^{N-1} \sum_{j=-N}^{N-1} \exp\left[-\widetilde{j} \cdot K_{sc} \cdot \left\{(\pi/N)(k \cdot i + m \cdot j) - \Delta\varphi_{i,j}^{x,y}\right\}\right], \tag{3}$$

where $\widetilde{j}$ is the imaginary unit and $K_{sc}$ is the scale factor that determines the size of the analyzed area within the far diffraction zone. Indices $i, j$ determine positions of the displayed elements of the phase screen, and indices $k, m$ correspond to positions of the analyzed points in the far diffraction zone. Expression (3) essentially describes the two-dimensional discrete Fourier transform of the transmission function of the synthesized phase screen. In the case of instrumental implementation of the considered technique, the far diffraction zone is realized in the rear focal plane of the Fourier transforming lens (see Figure 1). The scale factor $K_{sc}$ is determined by the following relationship between the focal length $F$ of the lens, the wavelength $\lambda$ of the readout beam, the linear size $\Delta$ of the phase screen elements, and the pixel size $\Delta_r$ in the readout plane:

$$K_{sc} = \frac{\Delta \, \Delta_r}{F\lambda} \tag{4}$$

Increasing the scale factor results in a finer-scale ("panoramic") display of the diffraction pattern from the DNA-associated phase screen in the far diffraction zone. Note that in the case of modeling the diffraction mapping using a two-dimensional discrete Fourier transform (3), the maximum allowable value of the scale factor is 0.5. Large values lead to the manifestation of the aliasing effect [43] in the synthesized diffraction patterns.

In the framework of the considered technique, the polarization structure of the synthesized diffraction patterns can be analyzed and displayed in terms of the local values of the Stokes vector elements [44]. These local values are introduced as

$$\begin{cases} s_{k,m}^0 = \left(\left|E_{k,m}^x\right|^2 + \left|E_{k,m}^y\right|^2\right) \Big/ 2; \\ s_{k,m}^1 = \left(\left|E_{k,m}^x\right|^2 - \left|E_{k,m}^y\right|^2\right) \Big/ 2s_{k,m}^0; \\ s_{k,m}^2 = 2\left|E_{k,m}^x\right|\left|E_{k,m}^y\right| \cos(\delta_{k,m}) \Big/ 2s_{k,m}^0; \\ s_{k,m}^3 = 2\left|E_{k,m}^x\right|\left|E_{k,m}^y\right| \sin(\delta_{k,m}) \Big/ 2s_{k,m}^0. \end{cases} \tag{5}$$

The local value of the first element $s_{k,m}^0$ defines the total intensity of the diffraction pattern at the (k, m) point; $s_{k,m}^1$ characterizes the difference in intensities of the x- and y-components of diffracted light for this point. The third element is similar to the second element, but it is introduced for the coordinate system rotated at the angle of 45° with respect to the basic (x, y) coordinate system. Finally, the fourth element $s_{k,m}^3$ defines the intensity difference for the right- and left-circular polarized components of the diffracted light at the chosen point.

For further analysis, it is convenient to use the normalized local values $\widetilde{s}_{k,m}^{1,2,3} = s_{k,m}^{1,2,3}/s_{k,m}^0$ based on the fundamental relation for the Stokes vector elements of the polarized light $\left(s_{k,m}^0\right)^2 = \left(s_{k,m}^1\right)^2 + \left(s_{k,m}^2\right)^2 + \left(s_{k,m}^3\right)^2$. Note that the normalized local values vary from −1 to 1. The proximity of the $\widetilde{s}_{k,m}^{1,2,3}$ values to ±1 indicates the occurrence of an extreme polarization state at a given point, which is characterized by significant dominance of either a linear or a circular component.

Figures 2 and 3 display the examples of the modeled $\widetilde{s}_{k,m}^{1,2,3}$ distributions in the far diffraction zone in the case of polarization-based representation of the symbolic sequence corresponding to sequence # 1 from dataset # 2. Figure 2 corresponds to the case of the finest-scale (panoramic mapping, $K_{sc} = 0.5$) representation, whereas Figure 3 displays sufficiently more detailed images of modeled distributions in the near-axis area of the diffraction zone (detailed mapping, $K_{sc} = 0.1$).

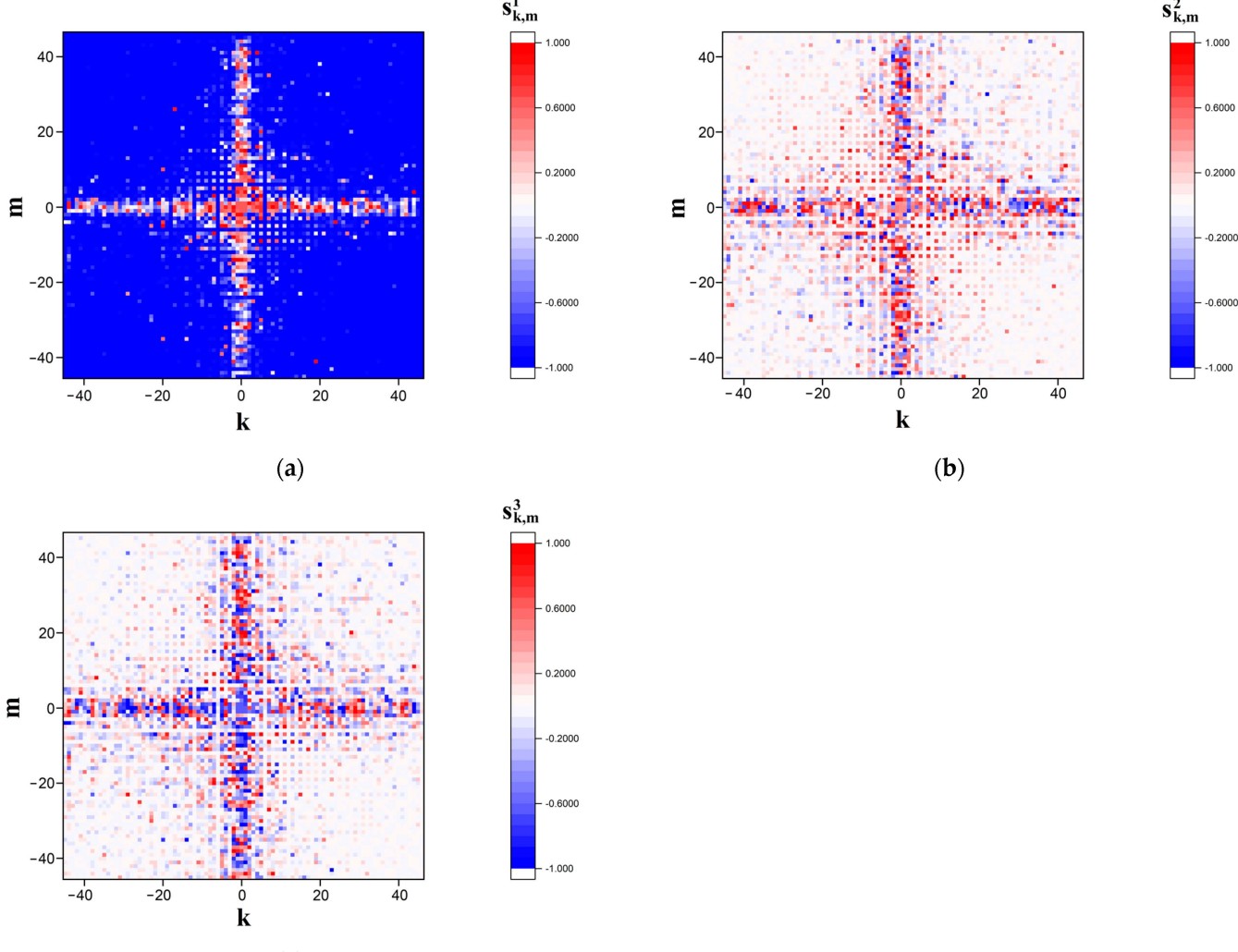

**Figure 2.** Modeled distributions of the Stokes vector components (**a**) $s_{k,m}^1$, (**b**) $s_{k,m}^2$, and (**c**) $s_{k,m}^3$ corresponding to sequence # 1 from dataset # 2 in the readout plane. The scale factor $K_{sc}$ is equal to 0.5 (the "panoramic" images).

Within the framework of the considered polarization encoding of nucleotide sequences, fingerprinting of a given sequence can be carried out by synthesizing a binary map of the local states of a close-to-circular polarization. For a given cutoff threshold $\widetilde{s}_{th}^3$, the synthesis algorithm has the following form:

$$
\begin{cases}
\widetilde{s}_{k,m}^3 > (<)\widetilde{s}_{th}^3 \rightarrow \widetilde{s}_{k,m}^{bin} = 1; \\
\widetilde{s}_{k,m}^3 < (>)\widetilde{s}_{th}^3 \rightarrow \widetilde{s}_{k,m}^{bin} = 0,
\end{cases}
\tag{6}
$$

where $\widetilde{s}_{k,m}^{bin}$ is the weight of the $(k, m)$ pixel in the synthesized fingerprinting map; case $>$ in the first formula corresponds to the mapping of extreme states close to the right-circular polarization, and the opposite case allows the selection of extreme local states with $s_{k,m}^3$

values approaching the left circular state. Figure 4a,b displays the fingerprints synthesized in this way for sequences ## 1 and 2 from dataset # 2 with a cutoff level of $-0.98$ (i.e., close-to-left circular states are displayed). The scale factor $K_{sc}$ is equal to 0.1 (the detailed imaging). High specificity of the synthesized binary maps is confirmed through their pixel-by-pixel logical multiplication (Figure 4c); note that when the number of unit pixels in the images a and b is 42, the number of the matching items is 18, indicating an appropriately high level of specificity. A more detailed analysis of the quantitative indicators of the specificity is presented in Section 3.

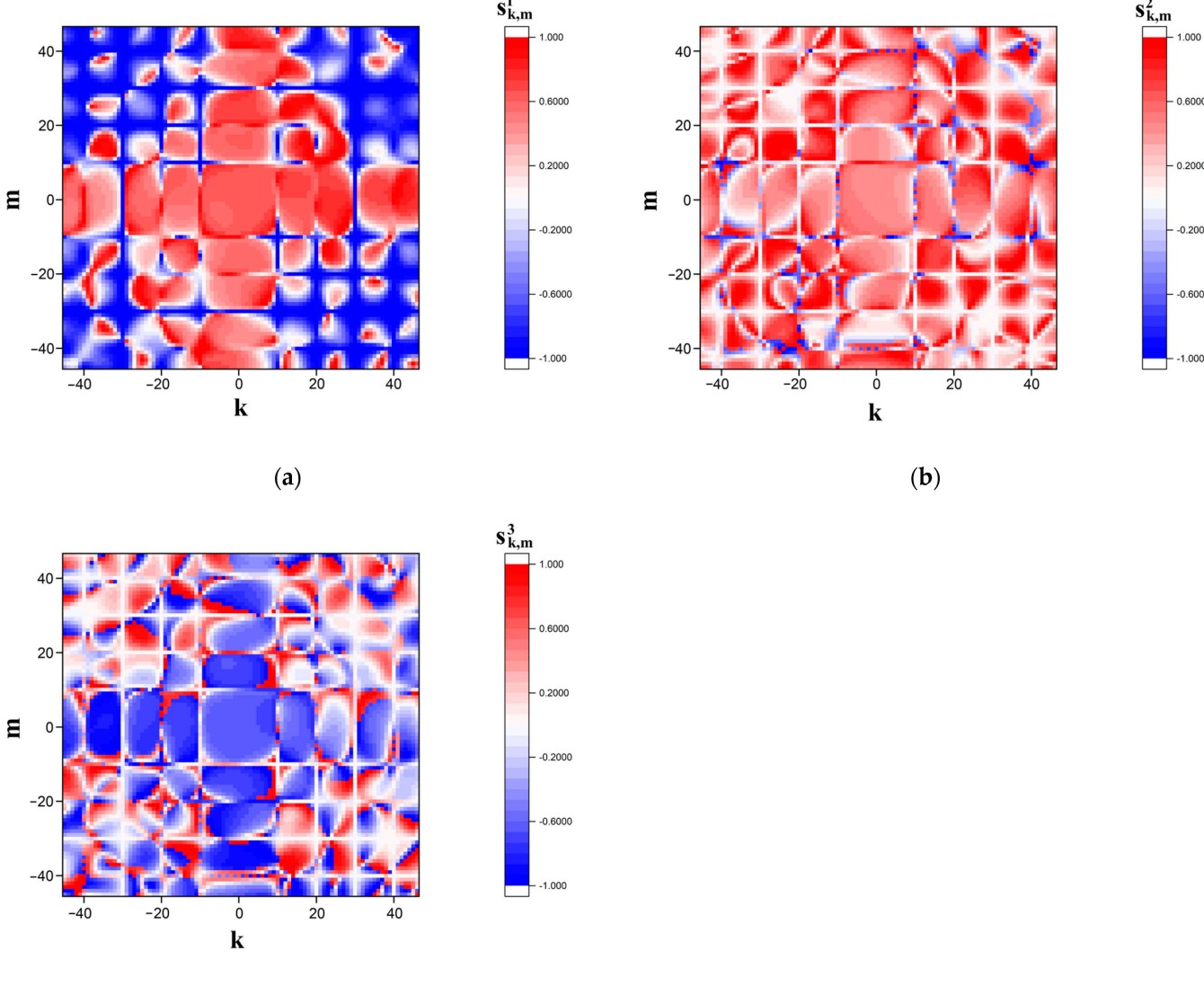

**Figure 3.** Modeled distributions of the Stokes vector components: (**a**) $s^1_{k,m}$; (**b**) $s^2_{k,m}$; (**c**) $s^3_{k,m}$; corresponding to sequence # 1 from dataset # 2 in the readout plane. The scale factor $K_{sc}$ is equal to 0.1 (the detailed images).

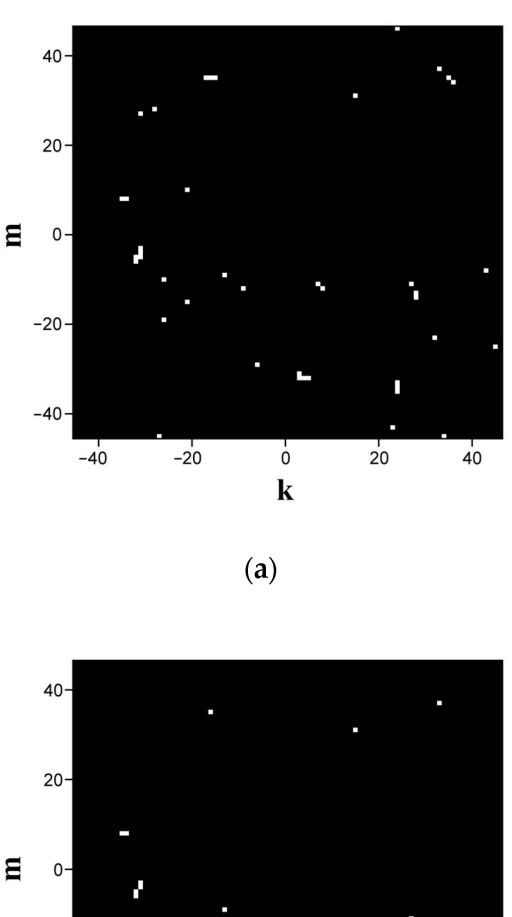

(a)

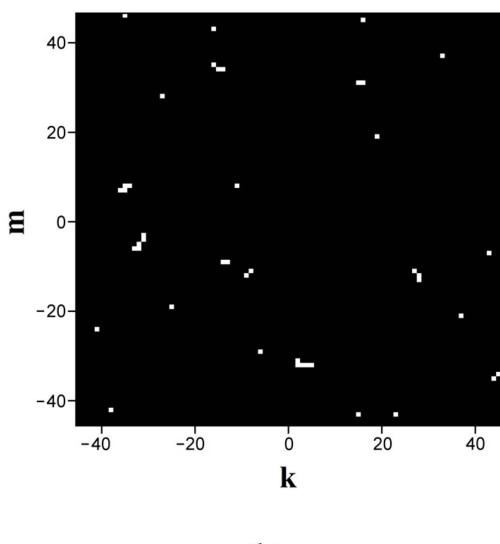

(b)

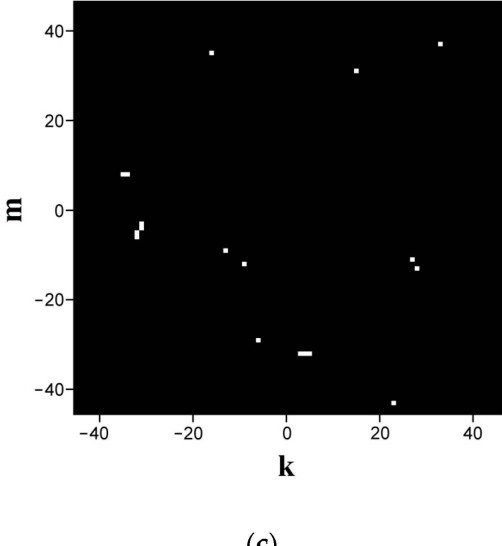

(c)

**Figure 4.** Polarization-based fingerprints for sequences (**a**) # 1 and (**b**) # 2; (**c**) panel displays the result of logical multiplication of the binary distributions (**a**,**b**).

### 2.3. CGR-Based Fingerprinting of Nucleotide Sequences: Basic Principles

Taking into account CGR mapping methodology (see, e.g., [8]), we can write the following set of expressions for the x- and y-coordinates of the imaging points corresponding to certain nucleotides in the displayed sequence. At the first step,

$$\begin{cases} x_1 = C_x^1/2; \\ y_1 = C_y^1/2, \end{cases} \tag{7}$$

where $C_x^1$, $C_y^1$ denote the x- and y-coordinates of the base node, corresponding to the first nucleotide in the sequence. The upper index denotes the position of the displayed nucleotide in the sequence. Positions of the base nodes ("nucleotide vertices", [8]) are given by the coordinate pairs $(0.0, 0.0)$, $(0.0, 1.0)$, $(1.0, 1.0)$, and $(1.0, 0.0)$. Regardless of the used basis (RY, MK, or WS), the A-node is always at the position $(0.0, 0.0)$. This offset of the starting point makes it possible to obtain more compact recurrence relations for the coordinates of the imaging points. For

example, when using the WS base for the CGR mapping, the node coordinates are as follows: $A \to (0.0, 0.0); C \to (0.0, 1.0); T \to (1.0, 1.0); G \to (1.0, 0.0)$. The RY base is characterized by the following associations: $A \to (0.0, 0.0); C \to (0.0, 1.0); G \to (1.0, 1.0); T \to (1.0, 0.0)$, and the MK base corresponds to $A \to (0.0, 0.0); G \to (0.0, 1.0); C \to (1.0, 1.0); T \to (1.0, 0.0)$.

At the second step of the CGR map synthesis,

$$\begin{cases} x_2 = x_1 + \left(C_x^2 - x_1\right)/2 = C_x^2/2 + C_x^1/4; \\ y_2 = y_1 + \left(C_y^2 - y_1\right)/2 = C_y^2/2 + C_y^1/4. \end{cases} \tag{8}$$

Accordingly, at the n-the step we obtain

$$\begin{cases} x_n = \sum\limits_{k=1}^{n} \dfrac{C_x^k}{2^{n-k+1}}; \\ y_n = \sum\limits_{k=1}^{n} \dfrac{C_y^k}{2^{n-k+1}}. \end{cases} \tag{9}$$

Figure 5 displays the recovered CGR maps in the RY, MK, and WS bases for the nucleotide sequence corresponding to sample # 1 from the dataset # 2.

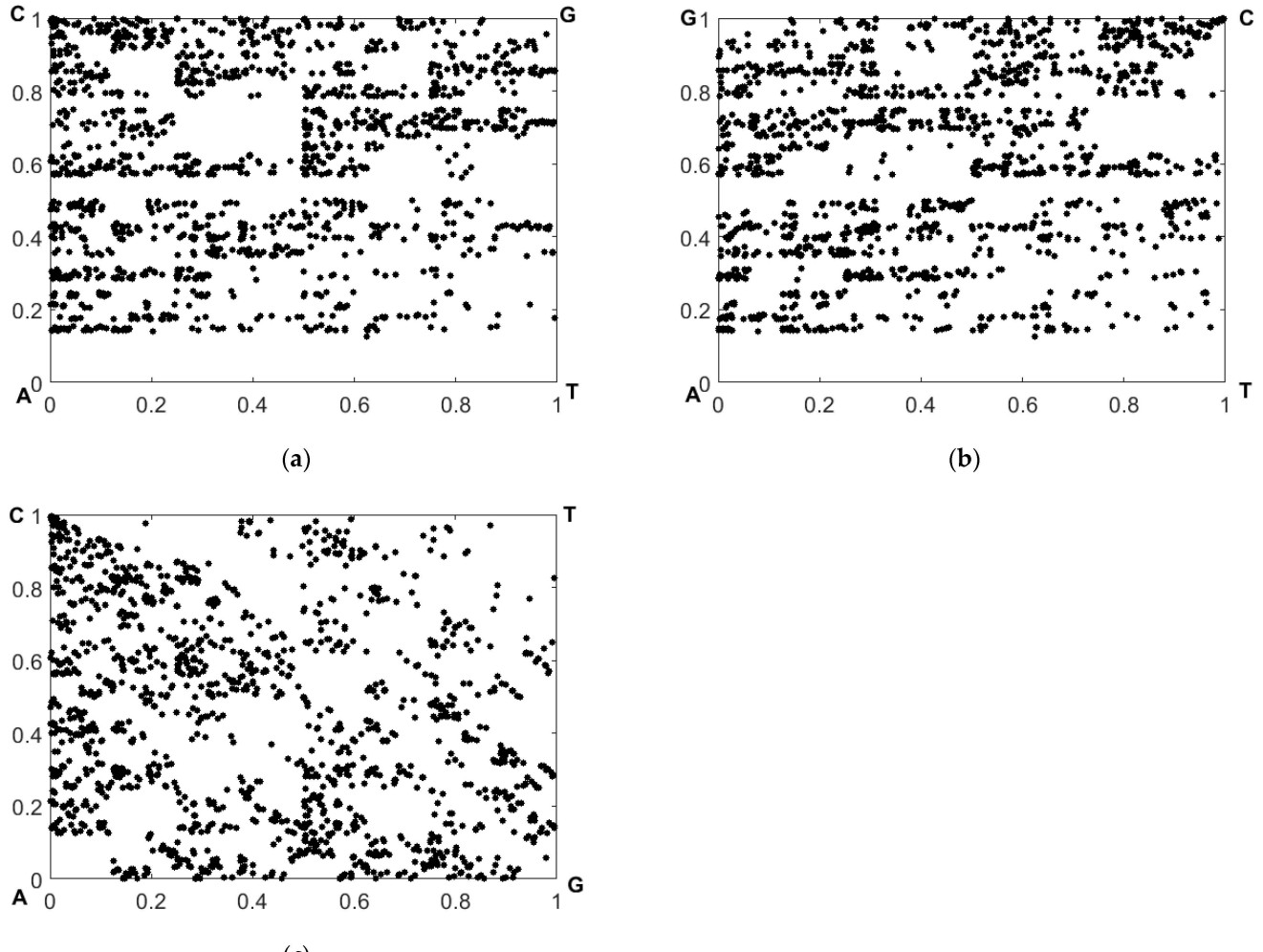

(a)

(b)

(c)

**Figure 5.** Recovered CGR maps in the (**a**) RY, (**b**) MK, and (**c**) WS bases in the case of sequence # 1 from dataset # 2.

The expressions (9) allow us to consider the decorrelation effect of the mutual positions of the imaging points in two CGR maps (reference and analyzed) resulting from a mutational substitution of one nucleotide in the sequence. The maximum differences in the coordinates $\Delta x = x - x'$; $\Delta y = y - y'$ occur for the imaging point corresponding to the substituted nucleotide, and gradually decay with the increase in the number $\tilde{n} = n_{cur} - n_s$ (here, $n_{cur}$ is the current position of the mapped nucleotide in the sequence, and $n_s$ is the position of the substituted nucleotide). It can be shown that this decay obeys the negative exponential law $\Delta x, \Delta y \propto 2^{-(\tilde{n}+1)} \propto \exp[-\ln 2(\tilde{n} - 1)]$. Figure 6 illustrates a gradual decay in $\Delta x, \Delta y$, with an increasing displacement of the representing nucleotides along the sequences with respect to the substitution position. The panels a and b display the CGR maps in the WS base for samples 1 and 2; panel c illustrates the effect of mismatching in the representing points due to occurrence of nucleotide replacement in sequence # 2 with respect to sequence # 1.

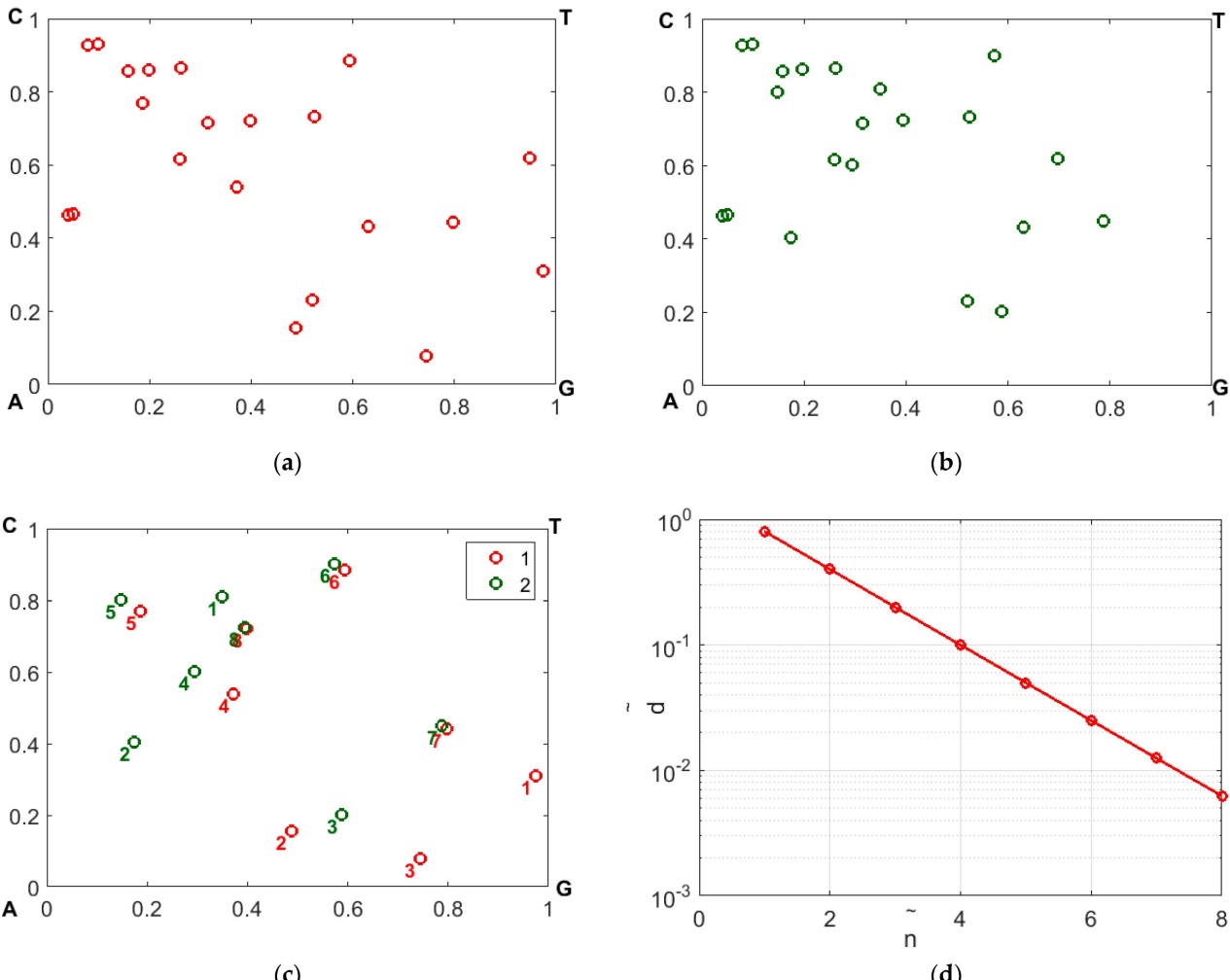

(a)  (b)

(c)  (d)

**Figure 6.** The effect of disappearance of differences in the coordinates of representing points in the reference and analyzed CGR maps with increasing displacement along the sequences from the substituted nucleotide: (**a**)—the fragment of CGR map in the WS base for the reference sequence (sequence # 1 from the dataset # 2); (**b**)—the same as in the panel (**a**) for the analyzed sequence (# 2); (**c**)—the combined sets of displaying points for the samples # 1 and # 2 inside the area of interest (numerical markers correspond to the displacement from the substituted nucleotide along the sequences); (**d**)—the distance between the mismatched representing points against the displacement from the substituted nucleotide in logarithmic coordinates.

The mismatched representative points are labeled by the values of $\widetilde{n}$. Panel d displays the distance $\widetilde{d} = \sqrt{(\Delta x)^2 + (\Delta y)^2}$ between the mismatched points in panel d depending on $\widetilde{n}$; the above-mentioned exponential decay is clearly seen.

The imaging points corresponding to certain nucleotides in the synthesized CGR maps are characterized by the real values of x and y coordinates in the (0.0; 1.0) intervals (i.e., their set is infinite). For numerical analysis of the correlation between the CGR maps of the reference sequence and the sequence with several substituted nucleotides, it is necessary to convert them to a matrix form, thereby reducing cardinality of the set of coordinate values of the representing points. Within the framework of this concept, the GCR map should be divided into $N^2$ sub-areas, and the presence and absence of imaging points in these sub-areas can be characterized by binary factors 1 and 0. Thus, finding a correlation between two similarly binarized CGR maps is reduced to an element-wise logical multiplication of two synthesized binary matrices, summation of the products dividing the result by the sum of the elements in the reference matrix. It is obvious that the level of fragmentation ("granulation") of the CGR map area, which makes it possible to minimize redundancy (more than one imaging point in a sub-area), is determined not only by the number of nucleotides in the analyzed fragment of the nucleotide sequence. Another key factor is existence of large-scale correlations of nucleotide positions in the analyzed sequences. These correlations are a manifestation of their fractal properties and, accordingly, are displayed in a significant heterogeneity of spatial distributions of the representing points in the CGR maps. A detailed analysis of this point is carried out in Section 3.

## 3. Results and Discussion

A quantitative comparison of the synthesized polarization- or CGR-based binary maps for the analyzed and reference symbol sequences can be carried out by calculating the correlation coefficients between them in accordance with the following expression:

$$R^{a,r} = \frac{\sum\limits_{i=1}^{N} \sum\limits_{j=1}^{N} a_{i,j} \times r_{i,j}}{\sum\limits_{j=1}^{N} r_{i,j}}, \tag{10}$$

where the designations $a_{i,j}$ and $r_{i,j}$ refer to the elements of the analyzed and reference maps that have the values 0 or 1. Thus, the numerator of the expression (10) determines the number of matching non-zero elements in the analyzed and reference maps, and the denominator determines the number of non-zero elements in the reference map.

### 3.1. Polarization-Based Fingerprinting

The correlation coefficients (10) between the polarization-based binary maps corresponding to the artificially generated symbol sequences from dataset # 2 are collected in Table 3. The synthesized maps correspond to spatial distributions of the normalized fourth component of the Stokes vector $\widetilde{s}_{k,m}^{3}$, and the threshold level $\widetilde{s}_{th}^{3}$ was chosen as $-0.95$ (the extreme states close to the left circular polarization were selected). The scale factor $K_{sc}$, equal to 0.1 (the detailed mapping), was applied.

The main diagonal in the matrix of correlation coefficients is highlighted in grey, and the cells with the unit correlation coefficients for different strains indicating identity of the symbol sequences are colored green. The following feature of the set of correlation coefficients should be mentioned, such as the distribution asymmetry of non-unit correlation coefficients relative to the main diagonal (for example, $R^{1,3} \neq R^{3,1}$). This feature is due to the differences in the number of unit elements in the synthesized binary maps for various strains.

**Table 3.** Correlation coefficients $R^{a,r}$ of polarization-based binary maps corresponding to various *L. monocytogenes*-based sequences from dataset # 2.

| No. of Sequence | 1 | 2 | 3 | 4 | 5 | 6 | 7 | 8 |
|---|---|---|---|---|---|---|---|---|
| 1 | 1.0 | 0.612 | 0.701 | 0.701 | 0.612 | 0.612 | 0.634 | 0.612 |
| 2 | 0.582 | 1.0 | 0.780 | 0.780 | 0.794 | 0.794 | 0.858 | 0.794 |
| 3 | 0.627 | 0.733 | 1.0 | 1.0 | 0.767 | 0.767 | 0.78 | 0.767 |
| 4 | 0.627 | 0.733 | 1.0 | 1.0 | 0.767 | 0.767 | 0.78 | 0.767 |
| 5 | 0.573 | 0.783 | 0.804 | 0.804 | 1.0 | 1.0 | 0.804 | 1.0 |
| 6 | 0.573 | 0.783 | 0.804 | 0.804 | 1.0 | 1.0 | 0.804 | 1.0 |
| 7 | 0.57 | 0.812 | 0.785 | 0.785 | 0.772 | 0.772 | 1.0 | 0.772 |
| 8 | 0.573 | 0.783 | 0.804 | 0.804 | 1.0 | 1.0 | 0.804 | 1.0 |

Some Remarks Regarding Polarization-Based Fingerprinting

Thus, binary identifiers of DNA-associated symbol sequences, created by their virtual polarization encoding and subsequent selection of extreme polarization states (preferably circular states with $s^3_{k,m}$ values close to $\pm 1$), characterize to an acceptable extent the uniqueness of the structure of the analyzed objects. The advantage of such "fingerprints" is the relatively small size of the corresponding black-and-white bitmap images. In particular, for the *Listeria*-associated sequences considered in this work, the number of binary elements in the identifiers is equal to $92 \times 92 = 8464$. Another advantage is the fairly high sensitivity of the synthesized binary patterns to small structural changes in sequences in the form of substitutions of a small number of nucleotides (for some quantitative estimates see Section 3.3).

*3.2. CGR-Based Fingerprinting*

When transforming a CGR map into a finite-dimensional matrix by dividing the map area into $N \times N$ equal-sized square fragments, the ideal case is that only one representative point hits (or misses) each fragment. Let us consider the estimate of the "admissible" degree of fragmentation, or the required number $N$ in the case of using the CGR map for the substitution-reduced sequences ## 1–8 (the dataset # 2). The "admissibility" criterion can be introduced based on the numerical analysis of the average redundancy indicators of the filling fragments with representative points. In particular, the ratio of the number of the $W$ cells with several points to the total number of nucleotides in the sequence ($N_n = 1587$) can be estimated depending on the $N/N_n$ ratio. Figure 7 displays the relationship between $W$ and $N/N_n$ in logarithmic coordinates for the CGR maps of sample # 2, which are recovered using RY, MK, and WS bases (markers 1–3). For comparison, a similar relationship was considered for the case of the uniform filling of the map area with 1588 points (marker # 4). The red and green lines serve as guides for the eye and indicate the trends in the decay of $W$, which are close to the power law in all the cases if $N/N_n > 0.1$. However, in the case of a uniform filling, the exponent approximates the expected value of 2, while for the RY, MK, and WS maps, its value is close to 1. This feature is due to the fractal nature of the distribution of representative points over the CGR maps, which leads to significantly non-uniform distributions of the local density points.

Figure 8 displays the calculated probability values $P_z$ of random finding of cells with various numbers of representing points (from z = 1 to z > 3) depending on the level of the CGR map (RY base) fragmentation for the symbol sequence corresponding to strain #2. As in Figure 7, the fragmentation level is given by the ratio $N/N_n$. Setting $P_{z=1} \geq 0.95$ as the "admissibility" criterion, we admit that this ratio must be at least equal to $\approx 0.64$ (i.e., the minimum number of elements in a binarized CGR map must be at least $\approx 10^6$ or more). Note that this value is many times greater than the similar value for the polarization-based binary maps synthesized for the same sequences ($\approx 8.5 \times 10^3$).

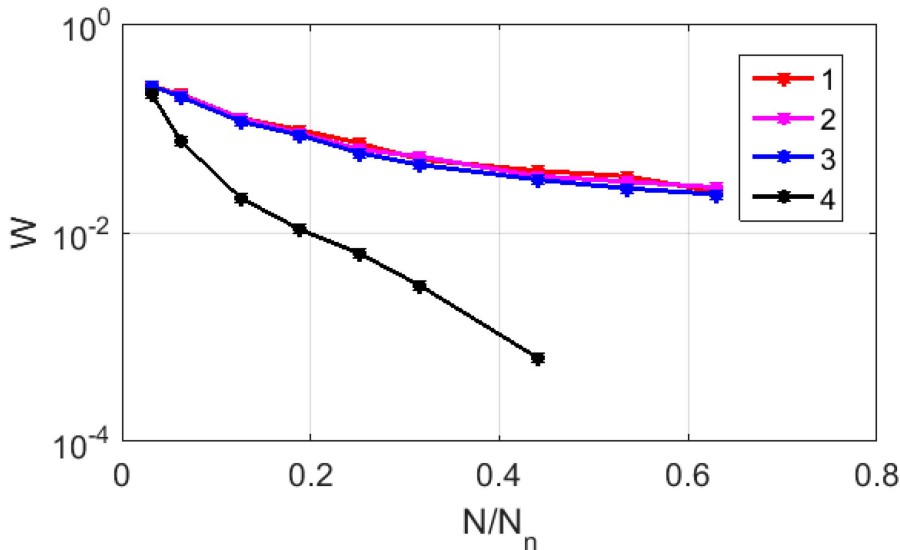

**Figure 7.** The ratio of the number of CGR map cells with several representing points to the total number of nucleotides in the sequence against the relative number of partitions $N/N_n$ in logarithmic coordinates; 1—the case of RY map (sequence # 2); 2—the case of MK map (sequence # 2); 3—the case of WS map (sequence # 2); 4—the case of uniform distribution of 1587 representing points across the area covered by CGR maps.

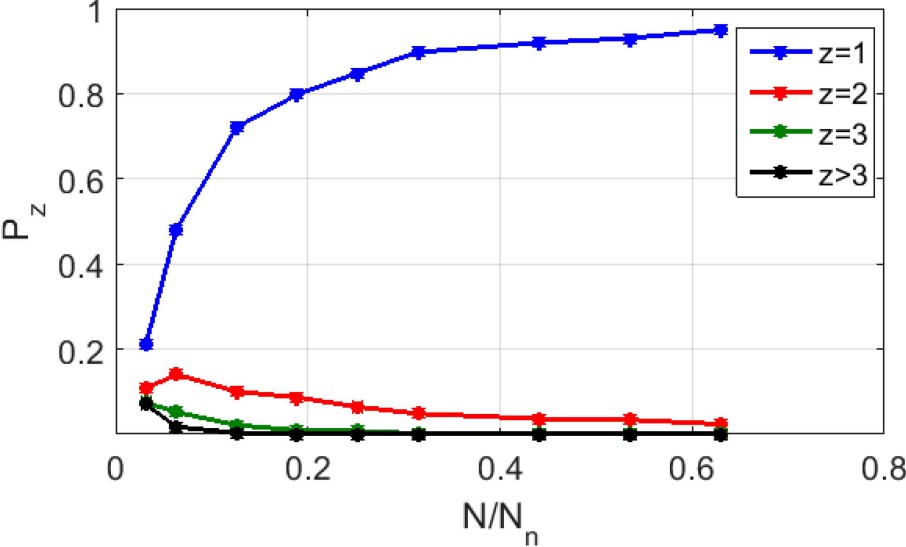

**Figure 8.** The modeled probability values $P_z$ of random finding of cells with various numbers of representing points (from z = 1 to z > 3) depending on the level of CGR map fragmentation; the used base is RY; the analyzed sequence is # 2 from the dataset # 2.

The analysis of the values of the correlation coefficients $R^{a,r}$ between the CGR-based binary maps for the sequences ## 1–8 with the reduced numbers of substitutions (the dataset # 2), depending on $N/N_n$, shows their rather weak dependence on the number of substitutions in the nucleotide sequences relative to the reference sequence. As an example, Figure 9 presents the values of the correlation coefficients against $N/N_n$ for the strain-associated CGR-based binary maps in the RY base. Note that the dependences $R^{a,r}(N/N_n)$ in the case of the MR and WS bases exhibit a similar behavior, shown by the close-to-unit values of even the criterion which is above the introduced "admissibility" and by the relatively large numbers of nucleotide substitutions. For comparison, the correlation

coefficients of the polarization-based binary maps show a significantly higher sensitivity to the changes in the structure of nucleotide sequences (see Table 3).

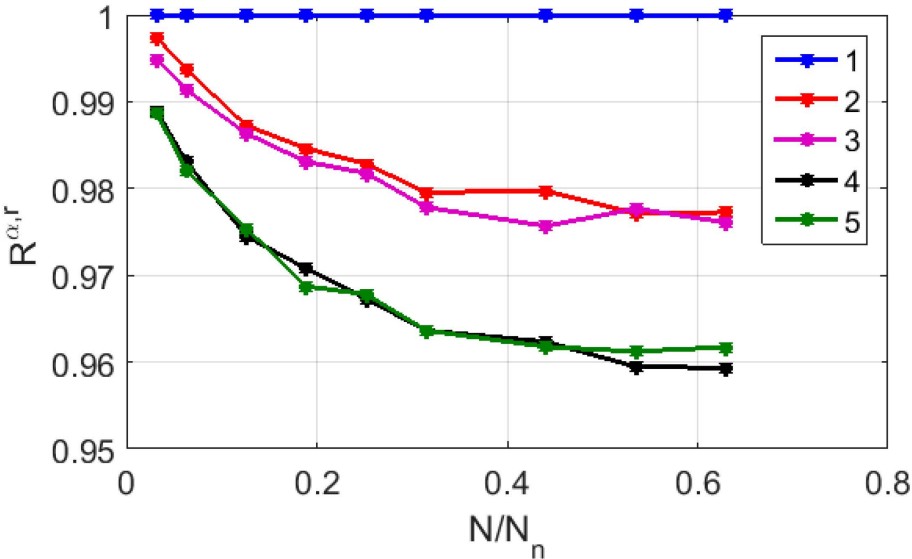

**Figure 9.** The correlation coefficients between the analyzed and reference (sequence # 1) fragmented CGR maps depending on the fragmentation level. The used base is RY; the compared pairs are: 1—sequence #1/sequence # 1 (number of substitutions is 0); 2—sequence #1/sequence # 7 (number of substitutions is 6); 3—sequence #1/sequence # 3 (number of substitutions is 7); 4—sequence #1/sequence # 2 (number of substitutions is 10); 5—sequence #1/sequence # 5 (number of substitutions is 11).

Some Remarks Regarding the CGR-Based Fingerprinting

Thus, compared to the polarization-based binary identifiers of DNA-associated symbol sequences, the CGR-based identifiers require the use of many times greater numbers of unit and zero elements to confidently establish the identity of or differences between analyzed sequences. However, even when the number of elements in binary identifiers is great, the correlation coefficient as an identification parameter in the second case turns out to be significantly less sensitive to small changes in the structure of sequences caused by substitutions of several nucleotides. For the analyzed sequences, the size of the binary matrices corresponding to the polarization-based identifiers is $92 \times 92$; in the case of CGR-based identifiers, this size is not less than $\approx 1000 \times 1000$. At the same time, the correlation coefficient $R^{a,r}$ when replacing, for example, 10 symbols in the reference sequence (sequence # 2 in dataset # 2 with reduced numbers of substitutions) decreases to 0.612 in the first case and falls to just $\approx 0.96$ in the second case.

*3.3. Robustness of $R^{a,r}$ Estimates for Polarization-Based Binary Identifiers*

Due to the quasi-random nature and variety of possible types of substitutions in the analyzed symbol sequences, the $R^{a,r}$ values will differ for different sequences with the same number of substitutions relative to the reference sequence. Regarding the polarization encoding of DNA-associated symbol sequences and synthesis of polarization-based binary identifiers $\left( \tilde{s}^{bin}_{k,m} \right)$, the effect of the variety of substitution types can be analyzed in terms of the transformation of a submatrix $\left( a_{ij} \right)_{2 \times 2}$ associated with the changed triplet. The following scenarios of $\left( a_{ij} \right)_{2 \times 2}$ transformation can be considered (note that the content and positions of symbols in the changed triplets are presented for illustrative purposes only).

(i) $\quad ACT \leftrightarrow \begin{pmatrix} 1 & 1 \\ 1 & 0 \end{pmatrix} \rightarrow ACG \leftrightarrow \begin{pmatrix} 1 & 1 \\ 0 & 1 \end{pmatrix}$, or $GCT \leftrightarrow \begin{pmatrix} 0 & 1 \\ 1 & 1 \end{pmatrix}$, or $AGT \leftrightarrow \begin{pmatrix} 1 & 0 \\ 1 & 1 \end{pmatrix}$;

(ii) $\quad ACT \leftrightarrow \begin{pmatrix} 1 & 1 \\ 1 & 0 \end{pmatrix} \rightarrow ACC \leftrightarrow \begin{pmatrix} 1 & 2 \\ 0 & 0 \end{pmatrix}$, or $CCT \leftrightarrow \begin{pmatrix} 0 & 2 \\ 1 & 0 \end{pmatrix}$, or $AAT \leftrightarrow \begin{pmatrix} 2 & 0 \\ 1 & 0 \end{pmatrix}$, etc.;

(iii) $\quad AAC \leftrightarrow \begin{pmatrix} 2 & 1 \\ 0 & 0 \end{pmatrix} \rightarrow AGC \leftrightarrow \begin{pmatrix} 1 & 1 \\ 0 & 1 \end{pmatrix}$, or $ATC \leftrightarrow \begin{pmatrix} 1 & 1 \\ 1 & 0 \end{pmatrix}$;

(iv) $\quad AAC \leftrightarrow \begin{pmatrix} 2 & 1 \\ 0 & 0 \end{pmatrix} \rightarrow ACC \leftrightarrow \begin{pmatrix} 1 & 2 \\ 0 & 0 \end{pmatrix}$;

(v) $\quad AAC \leftrightarrow \begin{pmatrix} 2 & 1 \\ 0 & 0 \end{pmatrix} \rightarrow AAA \leftrightarrow \begin{pmatrix} 3 & 0 \\ 0 & 0 \end{pmatrix}$;

(vi) $\quad AAA \leftrightarrow \begin{pmatrix} 3 & 0 \\ 0 & 0 \end{pmatrix} \rightarrow AAC \leftrightarrow \begin{pmatrix} 2 & 1 \\ 0 & 0 \end{pmatrix}$, or $AAT \leftrightarrow \begin{pmatrix} 2 & 0 \\ 1 & 0 \end{pmatrix}$, or $AAG \leftrightarrow \begin{pmatrix} 2 & 0 \\ 0 & 1 \end{pmatrix}$.

In the case of a single substitution to the reference sequence, the correlation coefficients $R^{a,r}$ of the corresponding binary identifiers will significantly differ from each other depending on the transformation scenario (i–vi). Table 4 presents the results of statistical modeling of the decorrelation between the binary identifiers of the reference and changed sequences in the case of a single substitution. The symbol sequence # 1 from the dataset # 2 was chosen as the reference object. The polarization-based binary identifiers $\left( \widetilde{s}_{k,m}^{bin} \right)$ were synthesized according to the procedure described above, with the following parameters: $K_{sc} = 0.1$; $\widetilde{s}_{th}^{3} = -0.95$. For the cases (i–vi), single replacements of symbols were repeatedly made in the corresponding triplets, selected in a random manner, and sets of 10 statistically independent $R^{a,r}$ values were accumulated in each case. The table displays the average values of the correlation coefficient with confidence intervals corresponding to a significance level of 0.9. It can be seen that the maximum decorrelation of binary identifiers with random single substitutions occurs in case v, while single substitutions in accordance with scenarios i and iv lead to minimal differences between the reference and analyzed identifiers.

**Table 4.** The modeled values of $R^{a,r}$ for various transformation scenarios in the case of a single substitution.

| Transformation Scenario | i | ii | iii | iv | v | vi |
|---|---|---|---|---|---|---|
| $R^{a,r}$ | $0.946 \pm 0.026$ | $0.740 \pm 0.043$ | $0.756 \pm 0.048$ | $0.946 \pm 0.027$ | $0.701 \pm 0.061$ | $0.708 \pm 0.093$ |

Note that substitutions corresponding to scenarios i and iv are associated only with permutations of the submatrix elements, while scenarios leading to maximum decorrelations under single substitutions are also associated with changes in the relationships between elements.

Figure 10 displays the pairwise-averaged correlation coefficients $\overline{R^{a,r}} = (R^{a,r} + R^{r,a})/2$ against the number of substitutions $N_s$ for both analyzed sets of sequences # 1 and # 2. The reason for using $\overline{R^{a,r}}$ is because the binary identifiers "a" and "r" can differ from each other in terms of the numbers of single states $\widetilde{s}_{k,m}^{bin} = 1$ (up to 10%), and averaging allows the influence of this factor to be reduced to a certain extent. The corresponding pairwise differences between the basic sequences (dataset # 1) and the artificially generated sequences (dataset # 2) are presented in the Supplementary Materials (Tables S3 and S4). Note that the number of plotted data items in Figure 10 is less than the number of sequence pairs due to the coincidence of some $\overline{R^{a,r}}$ values for the equal $N_s$ values.

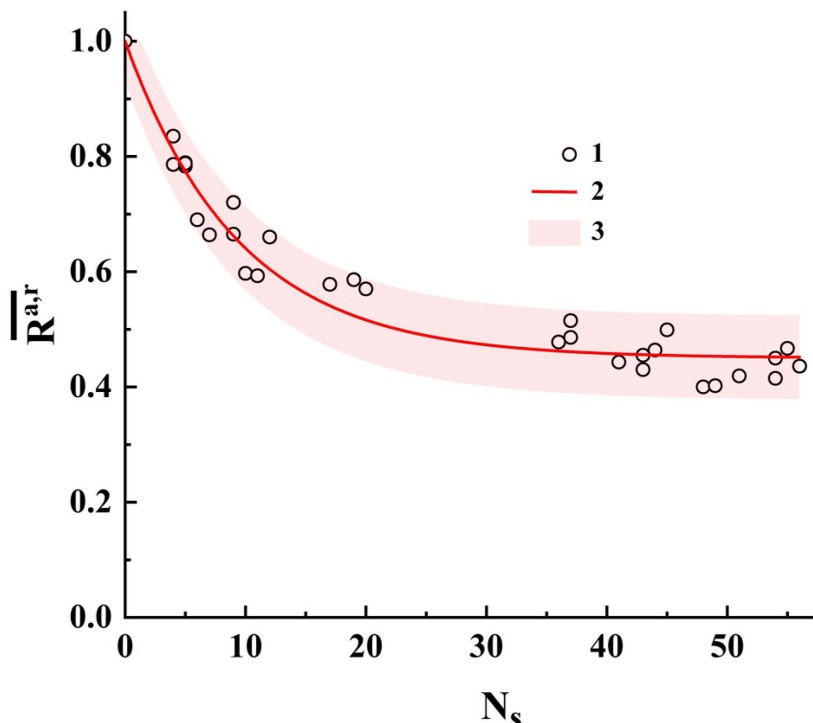

**Figure 10.** The pairwise-averaged correlation coefficients against the numbers of substitutions. 1—plot for datasets # 1 ($12 \leq N_s \leq 56$) and # 2 ($4 \leq N_s \leq 11$); 2—the fitting curve corresponding to the regression model (11); 3—the 95% prediction band.

The solid red line corresponds to the fitting dependence (regression model):

$$\overline{R^{a,r}} \approx 1 - K_1 \cdot [1 - \exp(-K_2 \cdot N_s)] \tag{11}$$

with $K_1 \approx 0.550 \pm 0.010$ and $K_2 \approx 0.106 \pm 0.005$. The fitting procedure was performed using OriginPro 2021 version 9.8.0.200 software. The adjusted R-squared value for this fitting is $\approx 0.949$; accordingly, we can conclude that the quality of the regression model (11) is acceptable. The pink-colored area corresponds to the 95% prediction band.

Note that the interpolated $\overline{R^{a,r}}$ value at $N_s = 1$ calculated using the regression model (11) is approximately equal to $0.944 \pm 0.032$, which is very close to the model values for the transformation scenarios i and iv. We can assume that these types of substitutions in nucleotide sequences are more likely to occur than other types. However, this assumption requires a more in-depth analysis and is considered an object for further research.

It is also interesting to consider the behavior of the regression model at large values of $N_s$, when the structures of the compared binary identifiers are almost uncorrelated. In this case where $\overline{R^{a,r}} \approx 0.45$, this value is determined by random coincidences of the positions of unit elements in two uncorrelated binary structures. The results of the performed analysis also confirm the previously made conclusion about the remarkably larger sensitivity of polarization-based binary identifiers to structural changes in the displayed sequences compared with CGR-based identifiers.

### 3.4. Potential Application of the Findings

A list of molecular DIVA assays has been proposed for a number of infectious animal diseases [45–52]. Although PCR (especially real-time PCR) is a rapid and sensitive technology with high specificity, providing good perspectives for the DIVA strategy, in some cases it does not suffice for the exact differentiation between vaccine and field strains. The example illustrating such problem is the PCR DIVA tests used for LSDV (lumpy skin disease virus), which could not properly assign recombinant vaccine-like viruses as either

a field or vaccine virus due to the variability in the target gene sequences [53]. Instead, sequence analysis of the whole genome or target genes may serve as a more accurate alternative for PCR tests. The subsequent sequence comparison may be performed with the use of both gene and genome sequence mapping technologies. For instance, this group of methods was applicable in phylogeny studies and genome similarity analysis [10,54–59]. The vast majority of them is based on CGR transform and shares common characteristics. When comparing the pair of sequences, the distance matrix of their complete CGR graphs must be calculated [55]. The CGR-based approaches have been successfully used even for interspecific discrimination of different bacterial and viral species including those that could not be classified by the traditional multiple sequence alignment methods based on the whole genome nucleotide sequences [60], particularly, when only incomplete genomic sequences are available [13]. However, a single-target gene analysis with smaller numbers of mismatches may be less fortunate.

Here, we consider another application of sequence visualization methods—with a perspective for DIVA strategy. In particular, two approaches to the two-dimensional binary mapping were used for the representation of the target gene nucleotide changes and differences between the vaccine and wild type bacterial strains on the model of *L. monocytogenes*, an important human and animal pathogen.

A comparative analysis of the sensitivity of polarization-encoded- and CGR-based binary maps to small deviations in the structure of the analyzed nucleotide sequences from the reference samples shows certain advantages of the polarization encoding approach. One of these advantages is associated with a significantly smaller size of a synthesized matrix of extreme polarization states compared to the binarized CGR map for the analyzed sequence. The small sizes of polarization-based binary identifiers are due to the essential properties of the discrete Fourier transform used in their synthesis.

Another advantage of the polarization encoding of DNA-associated symbol sequences is the significantly higher sensitivity of polarization-based "fingerprints" to the changes in the structure of the analyzed sequences due to the substitution of small numbers of nucleotides relative to the reference sequence. In fact, the obtained data proved that polarization-encoded binary map representation of the target artificially generated sequence dataset is comparable in accuracy to the classic, widely used alignment-based methods of sequence comparison [61]. Since this approach is more sensitive to single sequence mismatches, it can be used for pathogen interspecies discrimination in a more accurate way. It should be expected that a rather low sensitivity of binary representations to the structural changes is typical, not only in the case of the CGR-based approach, but also for other forms of representations based on the principle of iterative mapping in the four-point (A, C, T, and G) bases (in particular, 2D and 3D polylines).

The possibilities of the polarization encoding of nucleotide sequences are not limited to only the synthesis of their unique binary identifiers ("fingerprints"), and also include, for instance, the frequency analysis of the content of nucleotides in the sequences (the results of a pilot attempt of this kind of frequency analysis in relation to the SARS-CoV-2 virus are presented in [19]). In particular, we can consider the principle of constructing the triplet-associated submatrices in the RY, MR, and WS bases with a subsequent comparative analysis of the synthesized 2D binary maps as one of the possible directions for the further development of this technique. These feasible directions are a subject for further research activity.

The promising application of this strategy additionally to challenges in a general phylogeny, including phylogenetic and cluster analysis and evolutionary and global biodiversity research could be extended to a development of high-precision new generation point-of care diagnostic devices resulting in rapid and precise identification/differentiation of model *Listeria* and other pathogens and non-pathogens as well.

## 4. Conclusions

The results of the presented work demonstrated that the polarization-encoded approach was more potent in terms of the detection of minor nucleotide substitutions in the target artificially generated sequences than the CGR-based binary approach, and enabled an accurate differentiation between the vaccine and wild type *L. monocytogenes* strains. However, it was not as successful in the case of original sequences with a large number of nucleotide mismatches. The further development and testing of the described approach for the discrimination of various vaccine, field bacterial, and viral target gene sequences is needed and is of major interest for molecular DIVA assays, for which fast "in field" sequence analysis is favorable due to its feasibility in the case of the generation of short reads.

**Supplementary Materials:** The following supporting information can be downloaded at: https://www.mdpi.com/article/10.3390/cimb45120628/s1, Figure S1: Multiple sequence alignments of the original (a) and artificially generated sequences (b) *hly* gene nucleotide sequences of the vaccine *L. monocytogenes* strain AUF and seven wild type *L. monocytogenes* strains of zoonotic origin obtained using Multalin online service (http://multalin.toulouse.inra.fr/multalin/). The SNPs are colored in blue; Table S1: The homology level between the original AUF reference *hly* gene sequence and those derived from wild type *L. monocytogenes* strains of zoonotic origin (Figure S1a); Table S2: The homology level between the artificially generated sequences (Figure S1b); Table S3: The pairwise differences in nucleotides for the original *L. monocytogenes hly* gene sequences (Figure S1a; Table S1); Table S4: The pairwise differences in nucleotides for artificially generated sequences (Figure S1b; Table S2).

**Author Contributions:** Conceptualization, D.A.Z.; methodology, D.A.Z. and M.V.A.; software, D.A.Z. and M.V.A.; validation, D.A.Z., M.V.A., A.M.L., M.S.L. and V.A.F.; formal analysis, M.V.A.; investigation, M.V.A., A.M.L., M.S.L. and V.A.F.; resources, V.A.F.; data curation, M.V.A., M.S.L. and A.M.L.; writing—original draft preparation, D.A.Z. and A.M.L.; writing—review and editing, D.A.Z., A.M.L. and V.A.F.; visualization, M.V.A.; supervision, V.A.F.; project administration, V.A.F.; funding acquisition, V.A.F. All authors have read and agreed to the published version of the manuscript.

**Funding:** This research was funded by the Russian Science Foundation, grant number 22-16-00165.

**Institutional Review Board Statement:** Not applicable.

**Informed Consent Statement:** Not applicable.

**Data Availability Statement:** All analyzed nucleotide sequences were taken from open sources cited in the reference list.

**Conflicts of Interest:** The authors declare no conflict of interest.

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
