# Peer review of "Polarization- and Chaos-Game-Based Fingerprinting of Molecular Targets of Listeria Monocytogenes Vaccine and Fully Virulent Strains"

_cimb, doi:10.3390/cimb45120628_

Round 1
Reviewer 1 Report
Comments and Suggestions for Authors
The authors in the manuscript have used Polarization and chaos game based methods of fingerprinting of molecuar targets of Listeria monocytogenes vaccine and fully-virulent strains.
The authors have analyzed 8 strains and demonstrated that polarization-encoded approach is better than CGR based. However the whole manuscript discusses these two approaches but the details on the biological side of the study is missing. No conclusions or very little information on strain analysis results of Listeria monocytogenes has been made.
The manuscript appears confusing because of the same reason.
If the accuracy of the obtained results were verified?
Author Response
The authors are grateful for the valuable comments and suggestions of the reviewer, which are helpful for improvement of the manuscript quality. They are taken into account in the revised manuscript.
- “The authors in the manuscript have used Polarization and chaos game based methods of fingerprinting of molecuar targets of Listeria monocytogenes vaccine and fully-virulent strains. The authors have analyzed 8 strains and demonstrated that polarization-encoded approach is better than CGR based. However the whole manuscript discusses these two approaches but the details on the biological side of the study is missing. No conclusions or very little information on strain analysis results of Listeria monocytogenes has been made. The manuscript appears confusing because of the same reason.”
Our response:
The additional analysis of the analyzed original and artificially generated nucleotide sequences is presented in the introduced Supplementary Materials to this work.
“If the accuracy of the obtained results were verified?”
Our response:
The additional subsection 3.3 with the detailed consideration of the robustness of correlation coefficient evaluations and error estimates is introduced to the revised manuscript.
Reviewer 2 Report
Comments and Suggestions for Authors
Dear Authors,
Although the article contains very interesting ideas and an original approach, it still lacks comparisons with other articles in the specialized literature.
Although the article contains very interesting ideas and an original approach, it still lacks comparisons with other articles in the specialized literature. At least in the Results and discussions section, the article is inconsistent because it only analyzes the aspects dealt with by the authors, without comparisons and a broader approach.
Therefore, I think that the authors should analyze their own results in a broader vision, comparing them with the scientific data from the literature. For this purpose, the authors should introduce more references and discuss in detail the results obtained with more or less similar studies. This would increase the audience of interested readers, as well as the quality of the article.
Best regards
Comments on the Quality of English LanguageEnglish is fine, just a few typos errors.
Author Response
The authors are grateful for the valuable comments and suggestions, which are helpful for improvement of the manuscript quality.
“Although the article contains very interesting ideas and an original approach, it still lacks comparisons with other articles in the specialized literature. At least in the Results and discussions section, the article is inconsistent because it only analyzes the aspects dealt with by the authors, without comparisons and a broader approach.
Therefore, I think that the authors should analyze their own results in a broader vision, comparing them with the scientific data from the literature. For this purpose, the authors should introduce more references and discuss in detail the results obtained with more or less similar studies. This would increase the audience of interested readers, as well as the quality of the article.”
Our response:
Additional references were added to the revised manuscript (## 54-58 in Reference list). Regarding this point, we would like to provide some comments. The main goal of this work is a comparative analysis of two approaches to the binary representation of nucleotide sequences, one of which (the polarization-based binary representation) is being promoted by a team of authors. Results of a pilot study of the polarization approach using nucleotide sequences of the SARS-CoV-2 and African swine fewer viruses were previously reported in Refs. 18 and 19. Another approach (the CGR-based binary representation) uses an iterative algorithm for generating the sequence-associated ensembles of representation points in A, C, T, G bases. This algorithm is conceptually close to point-to-point procedures for generation of representation polylines and maps widely used in bioinformatics (see Refs. 3-17). The authors believe that Introduction, the cited references and consideration presented in Section 3 provide fair clarity about the purpose of the work and obtained results for the readers.
Reviewer 3 Report
Comments and Suggestions for Authors
The objective of the study was a comparative analysis of two approaches to the two-dimensional binary mapping of nucleotide sequences in terms of a unique correspondence between the structure of the synthesized maps and structure of the sequences (testing the “fingerprinting efficiency”) using for the objects of analysis the nucleotide sequences corresponding to the hly gene of the vaccine and various wild-type strains of the L. monocytogenes of zoonotic origin differing in the number of mutationally substituted nucleotides.
-The selection of the strains used in the study must be clarified and justified in detail. At the moment, the selection seems rather arbitrary.
-The model description must be separated in sub-sections to make it more reader friendly.
-Personally, I do not agree with the approach to mix results and discussion, but I do accept the authors view. However, I suggest that within each of sub-sections 3.1 and 3.2, a new paragraph titled ‘Comments’ (3.1.1 and 3.2.1) would be added to make clear the distinction and help the flow of reading.
-Some of the concluding comments are repetitive to the discussion previously and must be minimised.
-In conclusion, please add a new paragraph about the potential clinical benefits from the findings of the study.
Author Response
The authors are grateful for the valuable comments and suggestions, which are helpful for improvement of the manuscript quality. They are taken into account in the revised manuscript.
- “The selection of the strains used in the study must be clarified and justified in detail. At the moment, the selection seems rather arbitrary.”
Our response:
Actually, the current study is a part of the large research project devoted to the investigation of genomic DNA of Listeria. It has been recognized as the one of the severe pathogens for both human and animals (more information is presented in the Introduction section). The whole project is directed towards to the search of the approaches of Listeria virulent strains intraspecific discrimination following by the biodiversity investigation. Several our papers devoted to various aspects of this comprehensive study were published previously:
(i) Bespalova TY, Mikhaleva TV, Meshcheryakova NY, Kustikova OV, Matovic K, Dmitrić M, Zaitsev SS, Khizhnyakova MA, Feodorova VA. Novel Sequence Types of Listeria monocytogenes of Different Origin Obtained in the Republic of Serbia. Microorganisms. 2021 Jun 12;9(6):1289. doi: 10.3390/microorganisms9061289;
(ii) Zaitsev SS, Khizhnyakova MA, Feodorova VA. Retrospective Investigation of the Whole Genome of the Hypovirulent Listeria monocytogenes Strain of ST201, CC69, Lineage III, Isolated from a Piglet with Fatal Neurolisteriosis. Microorganisms. 2022 Jul 17;10(7):1442. doi: 10.3390/microorganisms10071442;
(iii) Khizhnyakova M.A., Zaitsev S.S., Ulianova O.V., Alexander Ulyanov, Ulyanov S.S., Feodorova V.A. Could the GB-speckles be used to study the evolution of Listeria monocytogenes? Optical Technologies for Biology and Medicine. – SPIE, 2022. – Т. 12192. – P. 134-140. DOI: https://doi.org/10.1117/12.2626298;
(iv) Kichemazova, N. V., Khizhnyakova, M. A., Lyapina, A. M., Kolosova, A. A. & Feodorova, V. A. Vaccine prophylaxis of listeriosis in farm animals. J. Veterinariya 3, 18-25 (2023).
Accordingly, a panel of Listeria-associated sequences and related artificially generated sequences with the reduced mismatches was considered in the current study to compare two methods of genetic «fingerprinting» for detection of nucleotide mismatches with the potential application for DIVA strategy. All of the strains used were sourced from farm animal with confirmed listeria infection. Also, only the strains with annotated whole genome sequences those are available in NCBI GenBank were included. These points are highlighted in the “Analyzed nucleotide sequences” subsection of “Materials and Methods”.
- “-The model description must be separated in sub-sections to make it more reader friendly.
-Personally, I do not agree with the approach to mix results and discussion, but I do accept the authors view. However, I suggest that within each of sub-sections 3.1 and 3.2, a new paragraph titled ‘Comments’ (3.1.1 and 3.2.1) would be added to make clear the distinction and help the flow of reading.”
Our response:
This was done; the additional subsections 3.1.1 and 3.2.1 were added to the revised manuscript.
- “Some of the concluding comments are repetitive to the discussion previously and must be minimised.”
Our response:
The “Conclusions” section was substantially shortened.
4 “In conclusion, please add a new paragraph about the potential clinical benefits from the findings of the study.”
Our response:
The brief description of the potential benefits and perspective applications of the findings obtained in the study was introduced as the additional subsection 3.4 of the revised manuscript.
Round 2
Reviewer 3 Report
Comments and Suggestions for Authors
All the comments have been answered. No more issues detected.
Author Response
The authors are grateful to the reviewer for evaluation of their work.